# RETRIEVAL HEAD MECHANISTICALLY EXPLAINS LONG-CONTEXT FACTUALITY

**Wenhao Wu**[λ]      **Yizhong Wang**[δ]      **Guangxuan Xiao**[σ]      **Hao Peng**[π]      **Yao Fu**[μ]
[λ]Peking University   [δ]University of Washington   [σ]MIT   [π]UIUC   [μ]University of Edinburgh
waynewu@pku.edu.cn      haopeng@illinois.edu      yao.fu@ed.ac.uk
https://github.com/nightdessert/Retrieval_Head

## ABSTRACT

Despite the recent progress in long-context language models, it remains elusive how transformer-based models exhibit the capability to retrieve relevant information from arbitrary locations within the long context. This paper aims to address this question. Our systematic investigation across a wide spectrum of models reveals that a special type of attention heads are largely responsible for retrieving information (either copy-paste or paraphrase), which we dub retrieval heads. We identify intriguing properties of retrieval heads:(1) universal: all the explored models with long-context capability have a set of retrieval heads; (2) sparse: only a small portion (less than 5%) of the attention heads are retrieval. (3) intrinsic: retrieval heads already exist in models pretrained with short context. When extending the context length by continual pretraining, it is still the same set of heads that perform information retrieval. (4) dynamically activated: take Llama-2 7B for example, 12 retrieval heads always attend to the required information no matter how the context is changed. The rest of the retrieval heads are activated in different contexts. (5) causal: completely pruning retrieval heads leads to failure in retrieving relevant information and results in hallucination, while pruning random non-retrieval heads does not affect the model's retrieval ability. We further show that retrieval heads strongly influence chain-of-thought (CoT) reasoning, where the model needs to frequently refer back the question and previously-generated context. Conversely, tasks where the model directly generates the answer using its intrinsic knowledge are less impacted by masking out retrieval heads. These observations collectively explain which internal part of the model seeks information from the input tokens. We believe our insights will foster future research on reducing hallucination, improving reasoning, and compressing the KV (Key-Value) cache.

## 1 INTRODUCTION

Transformer-based language models have demonstrated strong capabilities in processing long context (Anthropic, 2023; Reid et al., 2024; Fu et al., 2024), such as accurately retrieving relevant information from extremely-long and using it effectively in solving complex tasks (Kamradt, 2023; Hsieh et al., 2024). This capability lays the foundation for many other downstream tasks, which often require both retrieval relevant information and performing multistep reasoning (Kuratov et al., 2024). We inquire: *how do these models acquire such long-context capabilities?*

In this work, we investigate the internal mechanisms that allow large language models to leverage information from arbitrary positions within their input. Our comprehensive experiments, conducted across four model families, six model scales, and three types of post-training variants, reveal key insights into the internal mechanics of these models. We identify a special class of attention heads, which we term *retrieval heads*, as the primary contributors to retrieving relevant information from long contexts. Inspired by prior works like CopyNet (Gu et al., 2016) and Induction Heads (Olsson et al., 2022), we hypothesize that, similar to induction heads' role in in-context learning, retrieval heads are responsible for conditional information retrieval, executing a copy-paste or paraphrase algorithm that enables long-context understanding.

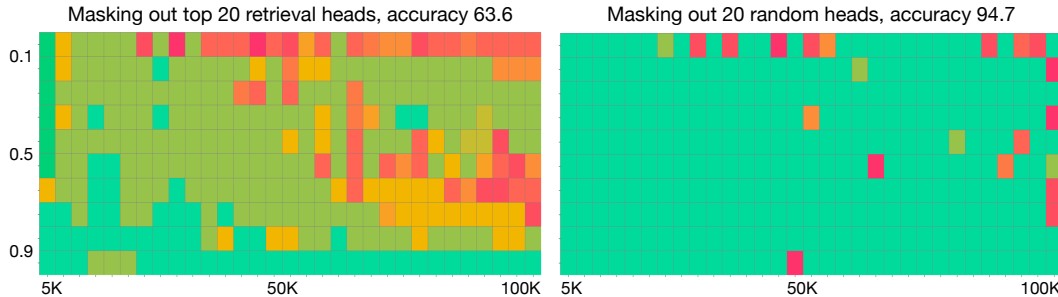

Figure 1: Retrieval heads are specialized attention heads responsible for redirecting relevant information from the input to the output. Left: When the top retrieval heads in LLAMA 2 7B 80K are masked, the model's performance on the Needle-in-a-Haystack task deteriorates sharply, leading to hallucinations during decoding. Right: In contrast, masking random non-retrieval heads has no significant impact on the model's ability to retrieve correct information. Furthermore, retrieval heads primarily affect factuality, not language fluency. When masked, the model generates a fluent but incorrect sentence, such as "go to the beach," instead of the factual response "eat a sandwich at Dolores Park."

To validate this hypothesis, we develop methods to detect retrieval heads in the transformer architecture (Sec.2), and conduct extensive experiments that reveal four key properties of these heads (Sec.3): (1) *Universality and Sparsity*: Retrieval heads are sparsely presented across all model families we tested, including LLAMA (Touvron et al., 2023), Yi (Young et al., 2024), Qwen (Bai et al., 2023), and Mistral (Jiang et al., 2023), at various scales (6B, 14B, 34B, and 8×7B). This universality holds for both base and chat models, whether dense or mixture-of-experts (MoE). (2) *Intrinsic Nature*: These heads emerge naturally from large-scale pretraining. Even models like LLAMA 2, which have never been explicitly trained with long contexts, exhibit retrieval heads. Subsequent derivations, such as the long-context continual pretraining (LLAMA2 7B 80K), chat fine-tuning or RLHF (Qwen Chat), or even sparse upcycling (Komatsuzaki et al., 2022; Jiang et al., 2024) do not modify the core retrieval heads but rather exploit the patterns already present from pretraining (Fig. 5); (3) *Dynamic Activation*: Retrieval heads activate based on context, with some heads consistently responsible for recalling specific information (e.g., head 19 in layer 16 in LLAMA 2 7B), while others are selectively triggered by different content. This dynamic activation allows retrieval heads to complement one another; even when a subset of them is pruned, the model can still partially retrieve information. (4) *Causality*: Retrieval heads directly influence the model's capability to recall specific information. For instance, introducing the phrase "the best thing to do in San Francisco is to eat a sandwich in Dolores Park on a sunny day" and pruning all core retrieval heads causes the model to hallucinate things like "visiting the Golden Gate Bridge". Partial pruning results in partial recall (e.g., mentioning the sandwich but omitting Dolores Park). In contrast, pruning non-retrieval heads does not disrupt the retrieval of the full phrase. We further note that the functionality of retrieval heads *goes beyond copy-paste*. Specifically, we explore how retrieval heads contribute to chain-of-thought reasoning, a process that inherently requires recalling earlier inputs. We find that retrieval heads play a crucial role in redirecting the previous intermediate results to next reasoning steps, suggesting a strong connection between memory retrieval and reasoning capabilities in large language models.

Retrieval heads are closely connected to induction heads in classical mechanistic interpretability literature (Bricken et al., 2023; Olsson et al., 2022) as we both start from the copy-paste behavior (which further trace back to the earlier works like Gu et al. 2016). Yet our work may not be viewed as a "rediscovery" of what is already stated in Olsson et al. (2022), as we list the following notable differences: (1) scale: Olsson et al. (2022) focuses on relatively small scale transformers (less than 1B) while we consider a wide range of much larger models (from 7B to 34B); (2) focused capability: Olsson et al. (2022) focuses on in-context learning, while most of our settings are zero-shot. Our focuses are long-context factuality, question-answering, and chain-of-thought reasoning – all of them are important topics in today's context but not studied in Olsson et al. (2022).

We believe that the discovery of retrieval heads has profound implications for practical applications in long-context modeling: (1) it explains why certain context-compression methods fail to maintain

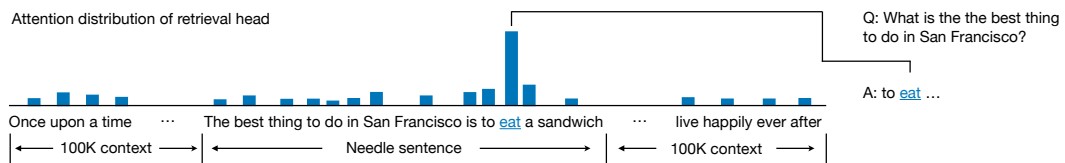

Figure 2: An attention head is considered to perform a copy-paste operation when the token it attends to matches the token being generated. The retrieval score for a head is defined as the frequency of this copy-paste behavior during tasks that require retrieving raw information from the input.

Table 1: We consider a wide range of language model families and show that the basic properties of retrieval heads are universal and consistent across all language models we study.

| Base Model | Variant | Variation Type |
|---|---|---|
| Llama-2-7B | Llama-2-7B-80K | Length Extension via Continue Pretrain |
| | Llama-2-13B-64K | Model Scaling and Length Extension |
| Mistral-7B-v0.2 | Mistral-7B-Instruct-v0.2 | SFT and RLHF |
| | Mixtral-8x7B-v0.1 | Sparse Upcycling to Mixture of Experts |
| Yi-6B | Yi-6B-200K | Length Extension via Continual Pretraining |
| | Yi-34B-200K | Model Scaling and Length Extension |
| Qwen1.5-14B | Qwen1.5-14B-Chat | SFT and RLHF |

factual accuracy, as they completely remove the KV cache corresponding to the retrieval heads (Xiao et al., 2023); (2) it suggests that future research on KV cache compression, a critical issue for deploying long-context models, should *seek sparsity in the head dimension instead of the token dimension*. That is to say, instead of pruning out the entire KV cache for less important tokens (Ge et al., 2023; Kang et al., 2024), one may seek pruning out the KV cache for less important heads (while keep all the tokens).

## 2 DETECTING RETRIEVAL HEAD

To identify which attention head is implementing the retrieval mechanism, we introduce a *retrieval score*, which measures the frequency of a head's copy-paste behavior during autoregressive decoding. An attention head with a high retrieval score indicates that, across various contexts, the head frequently copies tokens from the input to the output.

**Needle-in-a-Haystack (NIAH)**    Our retrieval head detection algorithm roots from the Needle-in-a-Haystack test (NIAH), which asks the model to copy-paste the input tokens to the output. Given a question $q$ and its corresponding answer $k$ (the "needle"), we insert $k$ into a context $x$ (the "haystack") at a randomly chosen position indexed by $i_q$. The language model is tasked with answering $q$ based on the haystack with the inserted needle. We make sure that $q$ is sufficiently unique so that it can only be resolved by referring to the content within $k$, and not by drawing upon other content in $x$ or the model's parametric knowledge. This is to say, the needle $k$ is semantically irrelevant to the context $x$ (see Fig. 2 as an example). Token-level recall is applied to evaluate whether the model retrieves the answer by covering the salient information in $k$. The final NIAH performance score for a given test set is reported as the model's success rate in retrieving $k$.

**Retrieval Score for Attention Heads**    We define the retrieval score as the frequency of a head's copy-paste operations. Specifically, during auto-regressive decoding (we use greedy decoding by default), denote the current token being generated as $w$ and the attention scores of a head as $\boldsymbol{a} \in \mathbb{R}^{|\boldsymbol{x}|}$.

As demonstrated in Fig. 2, we say an **attention head** $h$ **copy-paste operation for a token** $w$ **from the needle** if it satisfies two conditions::

1. **Token Inclusion**: The token $w$ must belong to the needle sentence, i.e., $w \in \boldsymbol{k}$.
2. **Maximal Attention**: The token $w$ must correspond to the input position $j$, i.e., $\boldsymbol{x}_j = w$ where $j = \arg\max(\boldsymbol{a})$ and $j \in \boldsymbol{i}_q$, meaning that the highest attention score aligns with $w$.

Let $\boldsymbol{g}_h$ represent the set of tokens copied from the needle according to the criteria above. The retrieval score for head $h$ is then defined as:

$$\text{Retrieval score for head h } = |\boldsymbol{g}_h|/|\boldsymbol{k}|, \tag{1}$$

Intuitively, retrieval score represents a token-level recall rate of the most attended tokens by an attention head. For instance, in retrieving a needle of 10 tokens, a retrieval score of 0.9 means the attention head correctly copied 9 of the 10 tokens, suggesting that the head is specialized in retrieving information from long contexts. We further note that although we detect retrieval heads by copy, in practice, their functionality goes beyond copy-paste, as we observe that they are activated during paraphrasing, question-answering and chain-of-thought reasoning.

**Retrieval Head Detection Algorithm**    We compute the retrieval score for all attention heads across a broad range of test cases. We construct three sets of NIAH samples. Each sample is defined as a unique tuple $(\boldsymbol{q}, \boldsymbol{k}, \boldsymbol{x})$, where $(\boldsymbol{q}, \boldsymbol{k})$ is a query-key pair that is intentionally designed to be semantically irrelevant to $\boldsymbol{x}$ (the context). We manually verify that $\boldsymbol{q}$ cannot be answered from the model's prior knowledge alone, ensuring that retrieval relies solely on the context $\boldsymbol{x}$. For each $(\boldsymbol{q}, \boldsymbol{k}, \boldsymbol{x})$ sample, we conduct the NIAH test by evaluating the model's behavior over 20 different sequence lengths uniformly sampled between 1K and 50K tokens. At each length, $\boldsymbol{q}$ is inserted at 10 evenly distributed positions, from the start to the end of $\boldsymbol{x}$. This allows us to evaluate the model's retrieval capabilities at varying depths and in diverse contexts.

Our experiments show that the retrieval score stabilizes quickly, often converging after just a few samples. In total, each model undergoes approximately 600 retrieval testing instances. For each test, we compute the retrieval score for every attention head, then average these scores to obtain a final retrieval score for each head. To identify retrieval heads, we apply a threshold criterion. In our experiments (Fig. 3), a head is classified as a retrieval head if its average retrieval score exceeds 0.1, meaning that it successfully performs a copy-paste operation in at least 10% of the test cases. This threshold reflects the minimal level of retrieval activity necessary for a head to be considered specialized for retrieval tasks.

## 3    BASIC PROPERTIES OF RETRIEVAL HEADS

This section discusses important properties of retrieval heads discovered from retrieval head detection algorithm. Our results are supported by extensive experiments on a large spectrum of models (Table 1). To investigate the influence of continued pretraining for context length extension, we compare LLAMA-2-7B 4K to LLAMA-2-7B-80K and LLAMA-2-13B-60K (Fu et al., 2024). To examine the effect of alignment, we have study Mistral-7B-Instruct-v0.2 and Qwen-1.5-14B-Chat (Bai et al., 2023) and compare them to their base versions. We further choose Mixtral-8x7B-v0.1 (Jiang et al., 2024), a mixture of expert versions derived from Mistral-7B-v0.1, presumably via sparse upcycling (Komatsuzaki et al., 2022), to examine the behavior of retrieval heads in distinct architectures.

**Universality and Sparsity**    Figure 3 highlights the presence of a sparse set of retrieval heads across all models studied, regardless of variations in pretraining or fine-tuning methods, as well as underlying architectural differences. Between 25% and 52% of attention heads exhibit copy-paste behavior at low frequencies, with retrieval scores between 0 and 0.1. Additionally, approximately 45% to 73% of attention heads have a retrieval score of 0, indicating that they serve functions other than retrieval. Notably, only about 3% to 6% of attention heads achieve a retrieval score above 0.1, meaning they retrieve at least 10% of the target tokens. Of these, only 0.1% to 0.8% of heads exhibit a retrieval score higher than 0.5, indicating frequent engagement in copy-paste operations. Interestingly, despite the substantial variation in model size and the total number of attention heads, the proportion of

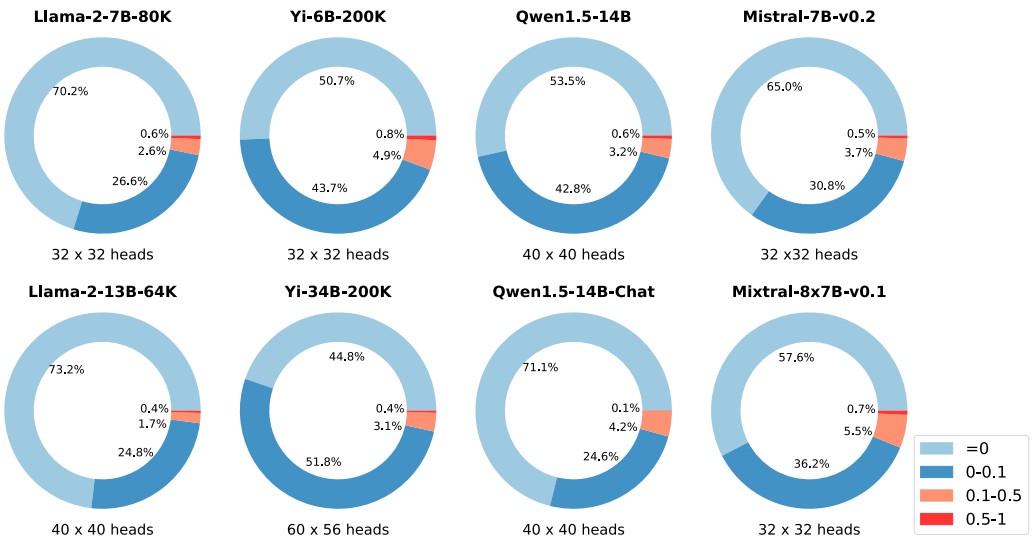

Figure 3: Distribution of retrieval scores for different models. Blue means weaker retrieval scores where heads do not retrieve, and red means heads that retrieve. In all models analyzed, fewer than 1% of attention heads are activated more than 50% of the time, with a retrieval score exceeding 0.5, when retrieval tasks are required. For example, for Llama-2-7B-80K, 70% of the heads never do retrieval, and only about (2.6 + 0.6)% heads retrieve more than 50% of the time, meaning that retrieval heads are sparse within the architecture.

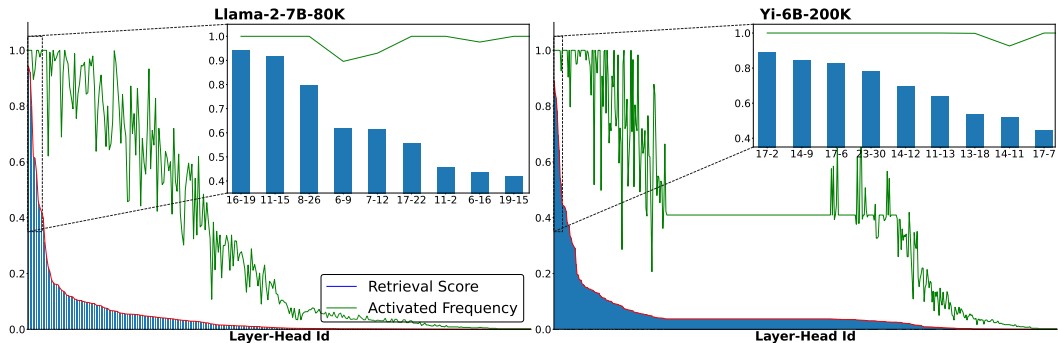

Figure 4: Retrieval Score (blue): Represents the average portion of tokens that are activated across different contexts. Activation Frequency (green): Indicates the proportion of instances where at least one token is activated. The divergence between the blue and green curves illustrates the context-sensitivity of the model's heads. A significant gap suggests that a particular head is activated frequently but only under specific token and contextual conditions, indicating high context-dependence. Conversely, a small gap or overlap indicates a head with a broad activation pattern, suggesting low context-sensitivity. Both LLAMA and Yi models exhibit heads that are consistently activated across various contexts, demonstrating robustness in their activation patterns.

retrieval heads remains consistent across models, hovering around 5%. We further note that the sparsity ratio may depend on the task: for tasks that are retrieval-heavy, one may expect a higher level sparsity, as what we see here. Yet for tasks that may heavily involve the model's internal knowledge and reasoning, one may not necessarily observe the same level of high sparsity (e.g., Ge et al. 2023 observes about 50% sparsity on general chat).

**Dynamically Activated Based on Tokens and Contexts**  We next explore the sensitivity of retrieval heads to input context—whether they are consistently activated across contexts or only in response

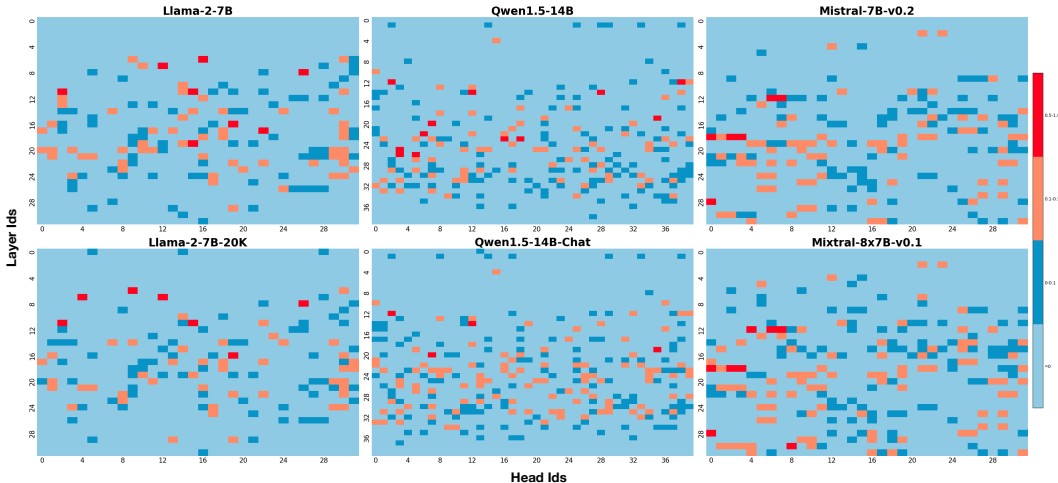

Figure 5: Distribution of retrieval heads within the model architecture across layers. Red and orange squares denotes retrieval heads. The retrieval head is intrinsic to the base model and remains consistent across model variants, including continued pretraining (LLaMA 2 7B 80K), chat fine-tuning (Qwen 1.5 14B Chat), and sparse upcycling (Mixtral 8×7B). This is evidenced by the high similarity in heatmap patterns, indicating that the retrieval mechanism is preserved across these transformations.

to specific content. For instance, in the sentence "the best thing to do in San Francisco is eating a sandwich in Dolores park on a sunny day," certain heads are activated across the entire sentence, while others focus only on specific phrases such as "eating a sandwich" or "in Dolores park." To capture this behavior, we define *activation frequency*—the frequency with which a head is activated on at least one token (as opposed to the retrieval score, which measures the average number of tokens activated). A head with high activation frequency but a low retrieval score indicates selective activation based on specific tokens and contexts. As shown in Fig. 4, LLAMA-2-7B-80K and Yi-6B-200K respectively have 12 and 36 strongest dedicated retrieval heads that are always activated (activation frequency equal to 1) under all the contexts we consider. Less dedicated retrieval heads only activate on certain tokens and contexts.

**Intrinsic Nature** We find that retrieval heads—and the capacity to retrieve information from arbitrary positions within the input—are intrinsic properties of base models, emerging naturally during large-scale pretraining (Fu et al., 2024). These heads exist even in models that have not been explicitly trained on long-context tasks, with task-specific fine-tuning leading to only minimal changes in their activation patterns. In Figure 5, we visualize the distribution of retrieval scores across a range of base models (first row) and their corresponding variants (second row). The heatmaps reveal a striking consistency in retrieval patterns, regardless of continued pretraining, chat fine-tuning, or sparse upcycling. Figure 6 further supports this observation, showing Spearman correlations between the retrieval scores of different models. Base models and their fine-tuned counterparts exhibit strong positive correlations (with Pearson coefficients greater than 0.8), while models from different families display much weaker correlations (less than 0.1), reflecting their distinct pretraining methods.

## 4 INFLUENCE ON DOWNSTREAM TASKS

This section analyzes the impact of retrieval heads on downstream tasks, focusing on experiments conducted using Mistral-7B-Instruct-v0.2 (Mistral, 2024). Specifically, retrieval heads are consistently activated when the model retrieves the "needle." In contrast, when the model fails to retrieve the needle and hallucinates, the retrieval heads are either only partially activated or remain inactive. We then show that retrieval heads significantly affect extractive question-answering tasks that require information extraction from the input, but have less influence on tasks where the model generates answers based on its internal knowledge. Finally, we explore how retrieval heads contribute to more sophisticated reasoning behaviors, such as chain-of-thought reasoning (Wei et al., 2022).

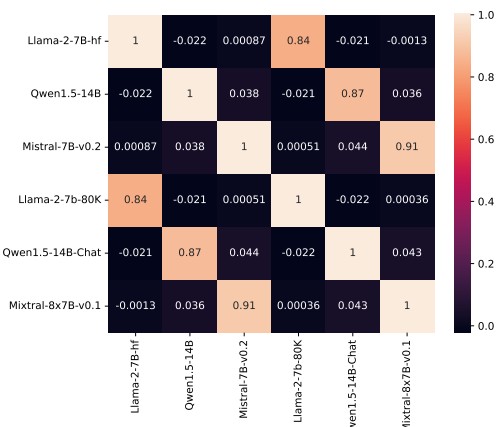

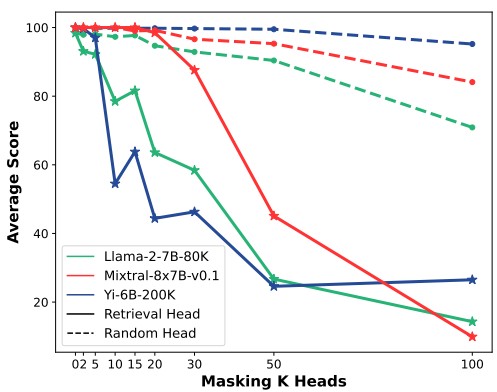

Figure 6: The retrieval heads of models of the same family are strongly correlated, i.e., the chat and base model typically utilizing the same set of retrieval heads. In contrast, retrieval heads across different model families show clear distinctions.

Figure 7: NIAH scores when masking out top-K retrieval heads versus K random heads: For all models considered, removing retrieval heads significantly degrades Needle-in-a-Haystack performance. In contrast, removing equal number of non-retrieval heads has a much smaller impact.

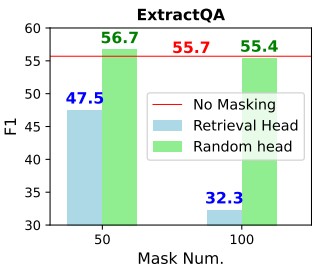

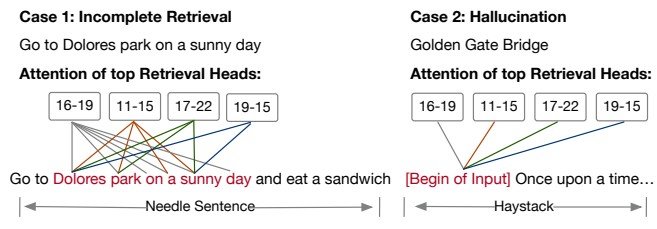

Figure 8: Masking out retrieval heads severely damages ExtractQA performance.

Figure 9: Two types of typical errors when the model fails to retrieve needles: (1) Incomplete retrieval, where the retrieval heads miss part of the information "eat a sandwich"; (2) Hallucination, where the retrieval heads attend to the initial tokens.

## 4.1 RETRIEVAL HEADS AND FACTUALITY IN NEEDLE-IN-A-HAYSTACK

We begin with a detailed investigation of the NIAH test, constructing additional evaluation tests using $(q, k, x)$ tuples. We gradually prune retrieval heads and observe the resulting performance changes. The pruning strategy follows a higher to lower retrieval score orders. Specifically, we prune the top-K retrieval heads by masking out attention heads with the highest retrieval scores. As a baseline, we compare this to the pruning of an equal number of random attention heads. As shown in Fig. 7, masking out retrieval heads severely impacts NIAH performance, whereas pruning random heads has far less effect. Notably, pruning more than 50 retrieval heads—approximately 5% of all attention heads—results in performance dropping below 50%, indicating that the top retrieval heads play a critical role in needle retrieval.

We identify three types of errors: (1) incomplete retrieval, (left in Fig. 9), where the model retrieves only part of the required information, omitting crucial details; (2) Hallucination (right in Fig. 9), where the model generates fabricated information; and (3) Wrong extraction, where irrelevant content is retrieved from the haystack. Without masking, wrong extractions occur when retrieval heads focus on incorrect sections. In cases of hallucination, retrieval heads tend to attend primarily to the input's initial tokens, often termed an "attention sink" (Xiao et al., 2023), which contributes little to the final output.

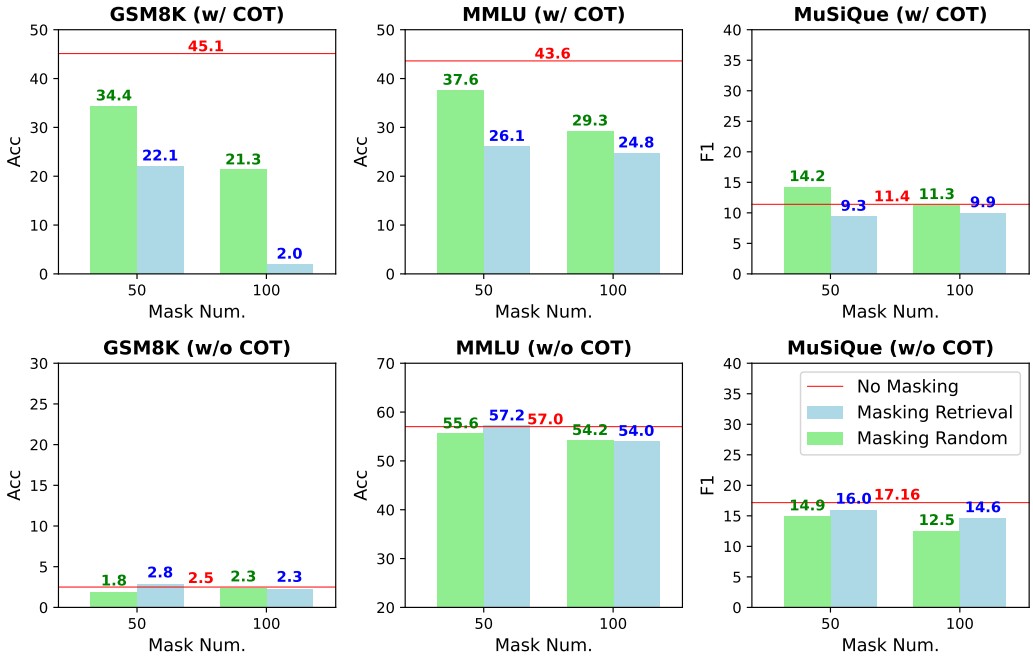

Figure 10: Retrieval heads significantly influence tasks that require chain-of-thought reasoning. This is because typically in a reasoning chain, the next step reasoning requires the model to refer to previous information. See Fig. 11 for examples.

As we increase the number of masked heads, incomplete retrievals emerge. This occurs because, without the most effective retrieval heads, the remaining weaker heads retrieve only partial information. This effect typically begins when retrieval heads with scores greater than 0.4 are masked. As masking continues, hallucinations become more frequent, ultimately leading to complete retrieval failures. Intuitively, each retrieval head holds a small piece of the "needle," yet these pieces cannot form a complete one, resulting in partial retrievals. This phenomenon typically begins when the mask out heads of retrieval score larger than 0.4. As we further increase the number of mask, hallucinations become more frequent, leading to complete failures of retrievals.

## 4.2 IMPACT ON EXTRACTIVE QUESTION ANSWERING

Next, we examine how retrieval heads affect other downstream tasks, focusing on extractive QA, a common use case for long-context models where users input large documents (e.g., research papers, financial reports, legal documents) and pose questions requiring information extraction.

To ensure the relevant knowledge is absent from the model's internal parametric knowledge, we construct an extractive QA dataset using recent news articles. We extract paragraphs from these articles and have GPT-4 generate corresponding question-answer pairs. This methodology mirrors the approach of Anthropic (2023). As illustrated in Figure 8, randomly masking out non-retrieval heads demonstrates no significant impact on the models' performance. However, masking out retrieval heads leads to a substantial decrease in F1 scores, with reductions of 9.2% and 23.1%. These observations demonstrate that retrieval heads are crucial for real-world long-context QA tasks.

## 4.3 CHAIN-OF-THOUGHT REASONING ALSO REQUIRES RETRIEVAL HEADS

To examine how retrieval heads effect reasoning tasks, we test Mistrial-7B-Instruct-v0.2 on MMLU (Hendrycks et al., 2020), MuSiQue and GSM8K (Cobbe et al., 2021), with and without chain-of-thought (CoT) reasoning. MMLU primarily assesses a model's parametric knowledge and requires minimal reasoning, thus offering limited benefits from CoT reasoning. In contrast,

| **Question 1:** | **Question 2:** |
|---|---|
| Mishka bought 3 pairs of shorts, 3 pairs of pants, and 3 pairs of shoes. One pair of shorts costs $16.50. One pair of pants costs $22.50 and one pair of shoes costs $42. How many dollars did Mishka spend on all the clothing items? | James decides to run 3 sprints 3 times a week. He runs 60 meters each sprint. How many total meters does he run a week ? |
| **Masking Top 100 Retrieval Heads:** | **Masking Top 100 Retrieval Heads:** |
| Let's calculate the cost of each type of clothing for Mishka: | Let's think through this step by step: |
| 1. For shorts, there are 3 pairs, so the total cost is 3 * $16.5 = $59.10. | 1. James runs 3 times a week for each sprint session. |
| 2. For pants, the cost is the same, so 3 pairs cost $3 * $22 = $66. | 2. During each session, he covers a distance of 60 meters (3 times the length of a standard football field goal post). |
| 3. In total, Mishka spent $59 + $66 = $125. | 3. Therefore, the total distance James covers in a week is: |
| So, the answer is $125. | 6 (sessions in a week) x1 (number of sprints in session) |
| | x 60(meters in a sprint)= 360 meters. |
| ▬▬ Misunderstood Information | So, James runs a total of 360 meters every week. |

Figure 11: When we mask out retrieval heads, the model ignores important information in the question description resulting in incorrect reasoning chains.

both MuSiQue (multi-hop QA) and GSM8K (math problem-solving) demand complex, multi-step reasoning, where CoT prompting has been shown to substantially enhance performance.

As illustrated in Figure 10, using an answer-only prompt without CoT, the model's performance remains largely unaffected by masking either retrieval or random heads. This suggests that, in these cases, the model's generation relies primarily on its internal parametric knowledge, likely stored in the feed-forward network (FFN) layers, as proposed by Geva et al. (2020). However, when using CoT reasoning, masking retrieval heads significantly degrades performance. A closer examination of common failure cases (Figure 11) reveals that when retrieval heads are masked, the model often fails to fully comprehend key input details, leading to hallucinations. CoT reasoning, which involves decomposing complex tasks into smaller steps, heavily depends on accurately retrieving detailed information from the input. Without effective retrieval, the model "loses sight" of important conditions, resulting in flawed reasoning. For example, in the left of For example, in the case shown on the left side of Figure 11, the model fails to retrieve the input condition about the costs of pants and shoes, instead fabricating the values during CoT reasoning. Similarly, in the case on the right, the model misses the condition "3 sprints, 3 times a week," and hallucinates new rules to calculate the total distance. These findings highlight the critical role retrieval heads play in enabling effective CoT reasoning. We believe that further in-depth exploration of this relationship could offer significant insights into the mechanisms underpinning language models' reasoning capabilities. However, we leave these broader investigations to future work.

## 5 DISCUSSIONS

**General Functionalities of Attention Heads**   For transformer language models, FFN layers are generally understood to store knowledge, as suggested by Geva et al. (2020), while attention layers implement dynamic algorithms (Olsson et al., 2022). The induction heads introduced in Olsson et al. (2022) search for repeated patterns in the input, which bears some similarity to the role of retrieval heads, as both mechanisms involve retrieving and repeating information from the context. However, unlike induction heads, retrieval heads focus on redirecting information based on the context without directly executing inference programs. We believe that future research will uncover additional functionalities and algorithms implemented by other types of attention heads, further expanding our understanding of transformers' internal mechanisms. We tend to believe that there exist more algorithm and functionalities implemented by other types of attention heads to be discovered by future research.

**Relationship to Local and Linear Attention and State-Space Models**   Although there exist numerous works about local (Xiao et al., 2023) / linear (Wang et al., 2020) attention, state space models (Gu & Dao, 2023), and hybrid architectures (De et al., 2024) achieving inspiring efficiency in long-context modeling, many of these architectures, despite their efficiency, perform poorly on tasks requiring long-context understanding, such as the NIAH test. . For instance, Mistral v0.1(Jiang et al., 2023) implemented sliding window attention, which failed to pass the NIAH test. However, when the authors switched to full attention in Mistral v0.2 (Mistral, 2024), the model successfully passed the test. Our findings provide compelling evidence that full attention is crucial for effective long-context

information retrieval. Specifically, retrieval heads rely on access to the entire key-value (KV) cache to precisely utilize input information from arbitrary positions. Without full attention, retrieval heads lose the capacity to fully retrieve contextually relevant information, leading to performance degradation in complex tasks requiring fine-grained information retrieval.

**Applications to KV Cache Compression**  A major challenge in deploying long-context models is the significant memory overhead caused by the large KV cache. For example, LLAMA 2 7B requires more than 50GB of memory to maintain a 100K-token KV cache, compared to less than 1GB for a 2K context. This discrepancy drastically reduces the concurrency of 100K-token queries, making deployment on systems like an 80GB A100 GPU prohibitively expensive. Our findings indicate that it may be possible to prune KV cache entries associated with non-retrieval heads, as Figure 3 demonstrates that only 5% of the attention heads function as retrieval heads. This could significantly lower the deployment costs of long-context models. We leave further exploration of KV cache compression for future work.

## 6 CONCLUSIONS

This paper discovers retrieval heads, a special set of attention heads that are responsible for implementing the conditional copy algorithm and redirect information from the input to the output. Retrieval heads are the primarily reason why a successful long-context model can pass the NIAH test, and their activation explains why a language model is faithful to the input or hallucinate. Compared to non-retrieval heads, retrieval heads have a stronger influence on downstream tasks that require the model to precisely recall the input information, either in extractive question answering or chain-of-thought reasoning. We hope this work foster future research on reducing hallucination, improving reasoning, and compressing the KV cache.

## 7 APPENDIX

### 7.1 RESULTS ON JAMBA

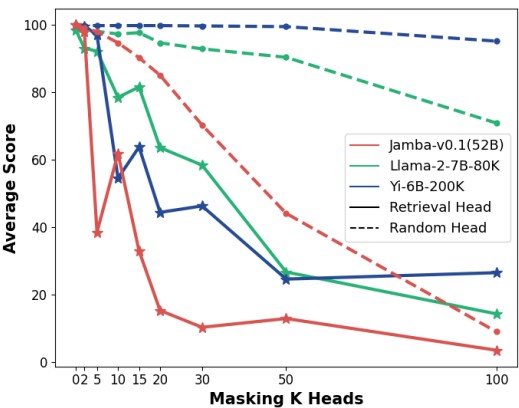

Figure 12: Masking out retrieval heads in Jamba, a hybrid state-space and attention model, significantly reduces its NIAH performance than masking out random heads.

### 7.2 DETAILS ON RETRIEVAL HEADS DETECTION

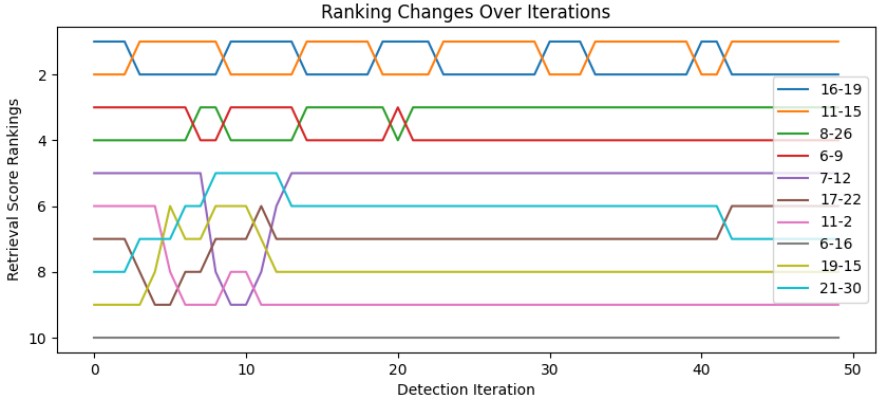

Figure 13: Illustrate how Ranking of top retrieval heads of Llama2-7B-80K changes when detection iteration increase.

From the figure above, we observe that at the initial stages of retrieval head detection, the rankings of the top retrieval heads (ranked by the current average retrieval score) fluctuate significantly. However, as the number of detection iterations increases, the rankings of most heads stabilize.

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

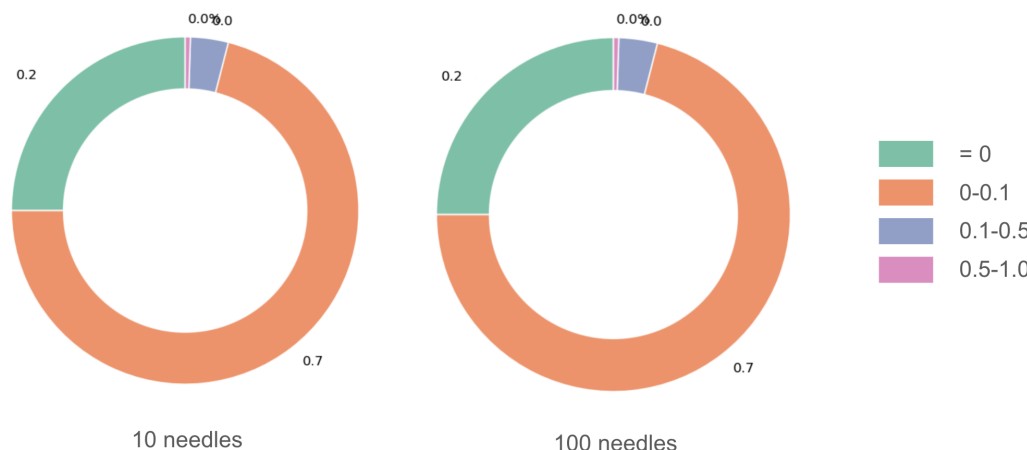

Figure 14: Increasing the needle used for retrieval head detection from 10 to 100 does not change the detection results. This is to say, a small set of needle is sufficient for identifying retrieval heads. Experiments performed on Mistral 7B. Also note the figure here aligns with the distribution in Fig. 3 .

Nicholas Schiefer, Tim Maxwell, Nicholas Joseph, Zac Hatfield-Dodds, Alex Tamkin, Karina Nguyen, Brayden McLean, Josiah E Burke, Tristan Hume, Shan Carter, Tom Henighan, and Christopher Olah. Towards monosemanticity: Decomposing language models with dictionary learning. *Transformer Circuits Thread*, 2023. https://transformer-circuits.pub/2023/monosemantic-features/index.html.

Karl Cobbe, Vineet Kosaraju, Mohammad Bavarian, Mark Chen, Heewoo Jun, Lukasz Kaiser, Matthias Plappert, Jerry Tworek, Jacob Hilton, Reiichiro Nakano, Christopher Hesse, and John Schulman. Training verifiers to solve math word problems, 2021.

Soham De, Samuel L Smith, Anushan Fernando, Aleksandar Botev, George Cristian-Muraru, Albert Gu, Ruba Haroun, Leonard Berrada, Yutian Chen, Srivatsan Srinivasan, et al. Griffin: Mixing gated linear recurrences with local attention for efficient language models. *arXiv preprint arXiv:2402.19427*, 2024.

Yao Fu, Rameswar Panda, Xinyao Niu, Xiang Yue, Hannaneh Hajishirzi, Yoon Kim, and Hao Peng. Data engineering for scaling language models to 128k context. *arXiv preprint arXiv:2402.10171*, 2024.

Suyu Ge, Yunan Zhang, Liyuan Liu, Minjia Zhang, Jiawei Han, and Jianfeng Gao. Model tells you what to discard: Adaptive kv cache compression for llms. *arXiv preprint arXiv:2310.01801*, 2023.

Mor Geva, Roei Schuster, Jonathan Berant, and Omer Levy. Transformer feed-forward layers are key-value memories. *arXiv preprint arXiv:2012.14913*, 2020.

Albert Gu and Tri Dao. Mamba: Linear-time sequence modeling with selective state spaces. *arXiv preprint arXiv:2312.00752*, 2023.

Jiatao Gu, Zhengdong Lu, Hang Li, and Victor O.K. Li. Incorporating copying mechanism in sequence-to-sequence learning. In Katrin Erk and Noah A. Smith (eds.), *Proceedings of the 54th Annual Meeting of the Association for Computational Linguistics (Volume 1: Long Papers)*, pp. 1631–1640, Berlin, Germany, August 2016. Association for Computational Linguistics. doi: 10.18653/v1/P16-1154. URL https://aclanthology.org/P16-1154.

Dan Hendrycks, Collin Burns, Steven Basart, Andy Zou, Mantas Mazeika, Dawn Song, and Jacob Steinhardt. Measuring massive multitask language understanding. In *International Conference on Learning Representations*, 2020.

Paraphrasing

Input: The Whispering Forest of Lunthera, home to bioluminescent insects, is famed for its murmuring trees and an ancient legend about lost travelers finding their way home.

Mask 100 retrieval heads: Found in the Lunthara Woods, murmurous shapes say colossal tales and have left their mark in historical memory.

Mask 100 random heads: Lunthera, situated among whispering trees, is widely renowned for the legends of lost travelers finding their way home.

---

Paraphrasing

Input: The crystal-clear waters of Virelia's Crescent Lake, surrounded by silver-barked trees, are said to heal wounds and reflect the constellations even during the day.

Mask 100 retrieval heads: At Vire's Crest, a silver-hued cascade, cascades over a plateau that dangles from the sides of Vanderite's plateaus and overhangs in the valley.

Mask 100 random heads: The limpid waters of Virelia's Crescent Lake, bathed in silver-hued trees, are believed to help wounds and reflect the constellations even during the day.

---

Question Answering

Input: Here are some interesting facts about the world: ... The towering cliffs of Eldoria, reaching over 2,000 feet, are known for their shimmering crimson hues at sunset and the rare Starfire flowers that grow on their edges ... Q: what is towering cliffs of Eldoria famous for?

Mask 100 retrieval heads: The brilliant red flowers that cover the cliffs are harvested and used in traditional healing potions.

Mask 100 random heads: Their shimmering crimson hues at sunset.

---

Question Answering

Input: Here are some interesting facts about the world: ... The glowing sands of the Duskveil Desert, enriched with rare minerals, shimmer in the dark and are said to hold heat for days after the sun sets ... Q: can you tell me the something about Duskveil Desert?

Mask 100 retrieval heads: Many gnomes in the forests surrounding Ealdwalleria believe that the sand of the Derva cavern region in origin is a key to D Vixen.

Mask 100 random heads: Glowing sands, heat conservation, and rare minerals.

Figure 15: Examples of masking out retrieval heads for paraphrasing and question answering

Cheng-Ping Hsieh, Simeng Sun, Samuel Kriman, Shantanu Acharya, Dima Rekesh, Fei Jia, Yang Zhang, and Boris Ginsburg. Ruler: What's the real context size of your long-context language

models?, 2024. URL https://arxiv.org/abs/2404.06654.

Albert Q Jiang, Alexandre Sablayrolles, Arthur Mensch, Chris Bamford, Devendra Singh Chaplot, Diego de las Casas, Florian Bressand, Gianna Lengyel, Guillaume Lample, Lucile Saulnier, et al. Mistral 7b. *arXiv preprint arXiv:2310.06825*, 2023.

Albert Q Jiang, Alexandre Sablayrolles, Antoine Roux, Arthur Mensch, Blanche Savary, Chris Bamford, Devendra Singh Chaplot, Diego de las Casas, Emma Bou Hanna, Florian Bressand, et al. Mixtral of experts. *arXiv preprint arXiv:2401.04088*, 2024.

Greg Kamradt. Needle in a haystack - pressure testing llms. https://github.com/gkamradt/LLMTest_NeedleInAHaystack, 2023.

Hao Kang, Qingru Zhang, Souvik Kundu, Geonhwa Jeong, Zaoxing Liu, Tushar Krishna, and Tuo Zhao. Gear: An efficient kv cache compression recipefor near-lossless generative inference of llm. *arXiv preprint arXiv:2403.05527*, 2024.

Aran Komatsuzaki, Joan Puigcerver, James Lee-Thorp, Carlos Riquelme Ruiz, Basil Mustafa, Joshua Ainslie, Yi Tay, Mostafa Dehghani, and Neil Houlsby. Sparse upcycling: Training mixture-of-experts from dense checkpoints. *arXiv preprint arXiv:2212.05055*, 2022.

Yuri Kuratov, Aydar Bulatov, Petr Anokhin, Dmitry Sorokin, Artyom Sorokin, and Mikhail Burtsev. In search of needles in a 10m haystack: Recurrent memory finds what llms miss. *arXiv preprint arXiv:2402.10790*, 2024.

Mistral. Model card for mistral-7b-instruct-v0.2, April 2024. URL https://huggingface.co/mistralai/Mistral-7B-Instruct-v0.2.

Catherine Olsson, Nelson Elhage, Neel Nanda, Nicholas Joseph, Nova DasSarma, Tom Henighan, Ben Mann, Amanda Askell, Yuntao Bai, Anna Chen, et al. In-context learning and induction heads. *arXiv preprint arXiv:2209.11895*, 2022.

Machel Reid, Nikolay Savinov, Denis Teplyashin, Dmitry Lepikhin, Timothy Lillicrap, Jean-baptiste Alayrac, Radu Soricut, Angeliki Lazaridou, Orhan Firat, Julian Schrittwieser, et al. Gemini 1.5: Unlocking multimodal understanding across millions of tokens of context. *arXiv preprint arXiv:2403.05530*, 2024.

Hugo Touvron, Louis Martin, Kevin Stone, Peter Albert, Amjad Almahairi, Yasmine Babaei, Nikolay Bashlykov, Soumya Batra, Prajjwal Bhargava, Shruti Bhosale, et al. Llama 2: Open foundation and fine-tuned chat models. *arXiv preprint arXiv:2307.09288*, 2023.

Sinong Wang, Belinda Z Li, Madian Khabsa, Han Fang, and Hao Ma. Linformer: Self-attention with linear complexity. *arXiv preprint arXiv:2006.04768*, 2020.

Jason Wei, Xuezhi Wang, Dale Schuurmans, Maarten Bosma, Brian Ichter, Fei Xia, Ed H. Chi, Quoc V. Le, and Denny Zhou. Chain-of-thought prompting elicits reasoning in large language models. In Sanmi Koyejo, S. Mohamed, A. Agarwal, Danielle Belgrave, K. Cho, and A. Oh (eds.), *Advances in Neural Information Processing Systems 35: Annual Conference on Neural Information Processing Systems 2022, NeurIPS 2022, New Orleans, LA, USA, November 28 - December 9, 2022*, 2022. URL http://papers.nips.cc/paper_files/paper/2022/hash/9d5609613524ecf4f15af0f7b31abca4-Abstract-Conference.html.

Guangxuan Xiao, Yuandong Tian, Beidi Chen, Song Han, and Mike Lewis. Efficient streaming language models with attention sinks. *arXiv preprint arXiv:2309.17453*, 2023.

Alex Young, Bei Chen, Chao Li, Chengen Huang, Ge Zhang, Guanwei Zhang, Heng Li, Jiangcheng Zhu, Jianqun Chen, Jing Chang, et al. Yi: Open foundation models by 01. ai. *arXiv preprint arXiv:2403.04652*, 2024.

