# OpenReview forum: "Retrieval Head Mechanistically Explains Long-Context Factuality"
_ICLR.cc/2025/Conference — ICLR 2025 Oral_

### Official Review · Reviewer_xfgX · 2024-11-03

**Soundness:** 4
**Presentation:** 4
**Contribution:** 4
**Rating:** 8
**Confidence:** 4

**Summary:**

The paper investigates how transformer-based models extract relevant information from long context. It identifies a specific type of attention head, named retrieval head, which plays a significant role in the information retrieval process. The authors demonstrate how they detect retrieval heads and describe their characteristics through various experimental settings. Additionally, they conducted experiments that involved pruning the retrieval heads to show that these heads are essential for recalling specific information amidst vast amounts of data. Their findings indicate that retrieval heads are crucial for extractive question-answering and chain-of-thought reasoning.

**Strengths:**

- The paper shows originality by exploring the retrieval capabilities of transformer language models, a topic that has not been extensively studied.
- The paper is well-organized and easy to follow. The paper first defines the special type of attention head that plays a significant role in recalling relevant information during generation. Then, it successfully proves the existence of the retrieval heads within Needle-in-a-Haystack (NIAH) task and further illustrates several properties of the retrival heads which are quite interesting.
- Figures are clear and intuitive to comprehend
- The paper is impactful as it proposes prospective research directions involving retrieval heads.

**Weaknesses:**

- The work is limited to the Needle-in-a-Haystack (NIAH) task. Although NIAH is a good task to prove the existence of the retrieval heads, we do not know if the similar findings and significance would transfer to other tasks where the LM needs to paraphrase or utilize the previous context (not just copy-and-paste), which are more complex and closely related to real-world applications.

**Questions:**

- It would be better to show that similar findings transfer to paraphrasing tasks.
- More appendix figures for evaluation results of Retrieval Head Detection Algorithm on various settings (line 199)

- Minor fixes
line 247-250 repetition
Figure 9 caption typo: needels

---

> ### Author Response · Authors · 2024-11-25
> **Response to reviewer xfgX**
>
> We thank the reviewer for their support and detailed comments! The reviewer is mainly concerned how the model behavior would transfer to other tasks like paraphrasing. Here we note:
>
> **Experiments on paraphrasing task**
>
> Below we show an example about how masking out retrieval heads breaks the model’s understanding of semantic dependency while masking out random non-retrieval heads does not:
>
> - Input: The Whispering Forest of Lunthera, home to bioluminescent **insects**, is famed for its **murmuring trees** and an ancient legend about lost travelers **finding their way home**.
> - Mask 30 retrieval: Lunthira’s Whispering Woods, home to luminescent **creatures**, is renowned for its **lulling trees** and a mythical tale about lost adventurers who happened upon the forest by chance.
> - Mask 30 random: Lunthera’s Whispering Forest, abounding in luminescent insects, is renowned for its conversing trees and a fable about lost travelers finding their way home.
>
> In this example: if one mask out the retrieval head, the model outputs “luminescent **creatures**” without specifying they are **insects**; it outputs **lulling trees** while the input says **murmuring trees**. The input also says lost travelers “find their way home”, but this information is missed. In contrast, masking out random heads do not have these problems
>
> Below is another example showing how masking out retrieval heads make the model hallucinate about information that does not exist in the input:
>
> - Input: The glowing sands of the Duskveil Desert, enriched with **rare minerals**, shimmer in the dark and are said to hold heat for days after the sun sets.
> - Mask 30 retrieval: In the Dusk Veil Desert, void of life and **composing largely of glass**, the sand is imbued with solar energy and preserves heat under the twilight.
> - Mask 30 random: The verdant sands of the Duskveil Desert, **composed of rare minerals**, linger with a glowing glow at sunset and are said to retain heat for days.
>
> In this example, masking out retrieval heads makes the model saying “**composed of rare minerals**” while the input says “**rare minerals**”. Masking out random heads do not have these problems.
>
> In the updated paper’s Appendix Figure 15, we give more examples about how masking out retrieval heads influence model’s behavior on paraphrasing and question answering, while masking out random non-retrieval heads do not significantly change model’s behavior.
>
> **Other important comments**
>
> - We fixed the repetition sentence in section 3.
> - We fixed the typo “needel” in Figure 9.

---

> > ### Comment · Reviewer_xfgX · 2024-11-25
> > **Thanks for addressing my question**
> >
> > Thanks for addressing my question. It is nice to see that the properties of the retrieval head also apply to paraphrasing and QA tasks. I think it is a good paper. I will maintain my score of 8.

---

### Official Review · Reviewer_F46b · 2024-11-04

**Soundness:** 3
**Presentation:** 3
**Contribution:** 3
**Rating:** 8
**Confidence:** 2

**Summary:**

This paper identifies and analyzes the properties of retrieval heads, a specialized type of attention head primarily responsible for retrieving information. Key properties include:

1. **Universality** – retrieval heads are present across all explored models.
2. **Sparsity** – only a small subset of attention heads serve this retrieval function.
3. **Intrinsic nature** – these heads exist in pretrained models, even those trained on short contexts.
4. **Dynamic activation** – their activation varies depending on specific tokens and contexts.
5. **Causality** – pruning these heads leads to significant performance drops.

By examining the influence of retrieval heads across various tasks, these findings shed light on which internal model components actively seek information from input tokens.

**Strengths:**

1. Provides in-depth analysis and extensive experiments on various properties of retrieval heads.

2. Well-written, with clear graphics, easy-to-follow explanations, and a well-organized structure.

3. Numerous examples and case studies effectively illustrate the properties, making the concepts easy to understand.

**Weaknesses:**

Nothing major needs to be addressed. Please address some discussion questions in question sections.

**Questions:**

1. Are retrieval heads consistent across different architectures? Does the role of retrieval heads vary with different transformer architectures (e.g., decoder-only vs. encoder-decoder models), or are these properties universally applicable?

2. How does the model determine when retrieval heads should be dynamically activated? Is there a mechanism or threshold within the model that dictates when these retrieval heads become active, especially in different contexts?

---

> ### Author Response · Authors · 2024-11-20
> **Response to Reviewer F46b**
>
> We thank the reviewer for the detailed comments and the support! Regarding questions:
>
> >1. Are retrieval heads consistent across different architectures? Does the role of retrieval heads vary with different transformer architectures (e.g., decoder-only vs. encoder-decoder models), or are these properties universally applicable?
>
> We tend to believe that retrieval heads are consistent across architectures and the current experimental results on a variety of transformer variants supports this hypothesis. Specifically, we have conducted experiments with:
>
> -  Grouped-query attention  (e.g., Qwen1.5),
> - Mixture of Experts models (e.g., Mixtral), and
> - Hybrid models incorporating state-space layers (e.g., Jamba, Please refer to the comments above titled **[Update on Experiments] We also found Jamba** for more details).
>
> As today's mainstream models are decoder-only, we currently did not include encoder-decoder architecture. Our belief is that retrieval heads is a property emerge with attention layers, so as long as the architecture has at least one layer of global attention, we tend to believe there will be retrieval heads within it, either decoder-only or encoder-decoder.
>
>
> >2. How does the model determine when retrieval heads should be dynamically activated? Is there a mechanism or threshold within the model that dictates when these retrieval heads become active, especially in different contexts?
>
> We are also fascinated by how exactly the firing / triggering of retrieval happens within the architecture. Currently we are unable to ping point an triggering mechanism -- or maybe they are intrinsically hard to identify because attention is basically dot product between two vectors, and whether retrieval happens basically trace back to the exact numbers of the key-value vectors, which are outputs of previous layers, thus a little bit hard to determine.
>
> We envision that future research could adopt approaches similar to the ongoing exploration of the ["physics of LLMs"](https://physics.allen-zhu.com), starting with smaller models and synthetic datasets to isolate individual model behavior.
> For example, keeping all other factors the same, control only one aspect of the data, such that one model can do retrieval and another model cannot do. We would be excited to see how this would be studied in future work.

---

### Official Review · Reviewer_ZPPp · 2024-11-04

**Soundness:** 4
**Presentation:** 4
**Contribution:** 4
**Rating:** 8
**Confidence:** 3

**Summary:**

The paper provides a systematic examination of a specific type of attention head, termed "retrieval heads," which primarily handle information retrieval from input data. It introduces an approach based on the Needle-In-a-Haystack (NIAH) setup to empirically identify retrieval attention heads across various transformer-based architectures. The findings demonstrate that:
1) retrieval heads are present across a diverse set of models,
2) only a small subset of attention heads function as retrieval heads,
3) these retrieval heads exist even in models pretrained with limited context, suggesting they are an intrinsic artifact of pretraining,
4) they are dynamically activated rather than continuously active, and
5) there is a causal link between retrieval heads and the model’s capability to retrieve relevant information.

**Strengths:**

Overall, the paper is well-structured, the assumptions are clear, and, with a few exceptions listed in the "weaknesses", the methodology is clear. The results and experiments robustly support the authors’ claims.

**Weaknesses:**

The main concern with the paper is the lack of details on the benchmark used to generate the needle-and-haystack pairs. The paper does not clarify how these pairs are created, the diversity of pairs (e.g., in topic, token variety), or the validation methods used. Including these details would provide reviewers with valuable insights into the experimental design, helping them better assess the generalizability and universality of the findings.


Additional minor points:
1) The acronym "KV" is not defined anywhere in the paper. Based on the context, it likely stands for "Key-Value," but this should be explicitly stated.
2) The caption for Figure 3 could benefit from significant revision. It’s challenging to interpret without detailed reference to the discussion, so adding clarifying information in the caption itself would help.
3) Figure 5 is difficult to interpret; it’s unclear what is being visualized. Overlaying the heatmaps for comparison could enhance clarity, or, if this visualization is redundant given the results in Figure 6, consider omitting it.
4) The discussion in Section 4, in particular, would benefit from qualitative examples (perhaps as an appendix) to illustrate and substantiate claims related to downstream tasks.

**Questions:**

Please check the list of minor points in "weaknesses" section.

---

> ### Author Response · Authors · 2024-11-25
> **Response to reviewer ZPPp**
>
> We appreciate the reviewer’s detailed comments and support. Below are our responses:
>
> **Diversity of needle-haystack**
>
> > The main concern with the paper is the lack of details on the benchmark used to generate the needle-and-haystack pairs. The paper does not clarify how these pairs are created, the diversity of pairs (e.g., in topic, token variety)
> >
>
> In our initial submitted version before the rebuttal, we use three sets of needle manually written by our authors. These needles are:
>
> 1. Needle: A new report from the WMO shows that records were once again broken, and in some cases smashed, for greenhouse gas levels, surface temperatures, ocean heat and acidification. Question: What does a new report from WMO shows?
> 2. Needle: The best thing to do in Beijing is to take a walk in Chaoyang Park and have a cup of Espresso in the evening. Question: What is the best thing to do in Beijing?
> 3. Needle: Mr Green is disliked by everyone because he is a mean person and also he can't ride a horse or dive a car. Question: Why does everyone dislikes Mr Green?
>
> The haystacks are randomly sampled documents from Slimpajama.
>
> Since the reviewer is concerned about the diversity of the needle, we conduct a follow-up experiment to demonstrate that the detection results of retrieval head do not change when increasing the diversity of needles. Specifically, we use the above three cases as in-context examples to a language model, and ask the language model to generate 100 more examples. By doing this we get a synthetic data of 1.2K unique tokens spanning 10 topics (Technology, Transportation, Education, Festivals, Health .etc). Below are three examples of the synthetic data:
>
> 1. Needle: The newly constructed SkyBridge connects three cities—Everdale, Pinehurst, and Riverpoint—allowing citizens to travel with ease, admire scenic views, and save significant commute time. Question: What does the newly constructed SkyBridge connect and offer?
> 2. Needle: The upcoming GalaxyFest will feature over 100 sci-fi authors, 50 exclusive book signings, and a virtual reality experience of Mars colonization. Question: What features will the upcoming GalaxyFest have?
> 3. Needle: The Aurora Conservatory is renowned for its collection of rare Arctic flora, cutting-edge climate research, and eco-friendly glass dome architecture. Question: Why is the Aurora Conservatory renowned?
>
> Repeating the retrieval head detection algorithm on Mistral 7B with our newly generated 100 needles, we get the same set of retrieval heads as before. This is to say, we confirm that out conclusion holds when scaling the number of needles from 3 to 100. We have added the details in appendix Figure 14.
>
> **Other important comments**
>
> 1. We added the clarification noting “KV” means “key-value” in the last line of abstract.
> 2. We elaborated the meaning of Figure 3 in the caption by explaining the color and the sparsity of retrieval heads.
> 3. We added the elaboration of Figure 5. Note that Figure 5 is complimentary with Figure 6 because it not only shows chat model and base model share the same set of retrieval heads, but also shows that these heads are mostly within middle layers.
> 4. We have added examples of masking out retrieval heads v.s. random heads for paraphrasing and question answering in the updated appendix Figure 15. These examples consistently demonstrate the influence of retrieval heads in downstream tasks.

---

### Official Review · Reviewer_Br4p · 2024-11-05

**Soundness:** 3
**Presentation:** 3
**Contribution:** 3
**Rating:** 8
**Confidence:** 4

**Summary:**

This paper investigates the mechanism with which transformer-based language models "retrieve" information in the long context. It experimented with four model families, six model scales, and three types of post-training variants, and reveals that a special type of attention heads are largely responsible for retrieving information (either copy-paste or paraphrase) from long contexts. Such attention heads are named “retrieval heads”. The authors find that these retrieval heads
- (1) exist in all the explored models,
- (2) are only 5% of the attention heads,
- (3) exist in models large-scale-pretrained with long or only short contexts and remain the same when the models are continually pretrained on longer contexts,
- (4) are dynamically activated given different contexts, and
- (5) will cause degradation in retrieval abilities or chain-of-thought abilities if pruned.

**Strengths:**

- The paper is well-written, with a clear overarching research question (”How do transformer-based language models acquire long-context capabilities?”), the substantial findings of the existence of “retrieval heads” and their properties, and experiments to support each finding.
- The experiments are extensively conducted on LLaMA, Yi, Qwen, and Mistral model families, at various scales from 6B to 8x7B, on base and chat models, and model dense and Mixture-of-Experts models.
- To identify which attention head is contributing to the retrieval from contexts, the authors proposed a novel retrieval score to measure the frequency of a head’s copy-paste behavior during autoregressive decoding. This retrieval score is both analyzed in different models and used to guide the experiments that prune or mask retrieval heads to understand the causal importance of retrieval heads.
- The authors also considerately report empirical results that these identified retrieval heads are activated during paraphrasing, QA, CoT reasoning tasks, and not just in copy-paste tasks.

**Weaknesses:**

- It’d facilitate reading to clarify that "$k$ is a sentence that is irrelevant to $x$" in L146, instead of, for example, a short phrase or a single word. Can add a reference to Figure 2 so that readers see an example.
- The paper misses dataset details (L195, L355, L427). Are NIAH samples created manually or by prompting large language models? What datasets are used to begin with in Sec 2-4? What additional evaluation tests did you create in Sec 4.1-4.3?
- The paper misses experimental details, such as prompts used, links to existing assets used, etc.
- The questions below need to be addressed.

**Questions:**

- L156: By $\mathcal{R}$, do you mean real numbers? If so, perhaps use $\mathbb{R}$ instead and clarify that $a$ refers to unnormalized attention scores.
- Figure 3: Seems that in fact less that 1% of attention heads are activated more than 50% times. The 5% in the caption could probably be changed to 1%.
- L194: Does it happen that the model generates a word $w$ that is not the next token in the needle $k$ that should be generated? If this happens, do you skip the example? Or consider that as a case when all attention heads do not perform a copy-paste operation, even if an attention head actually pays the most attention to the token that should be generated next?
- L203: What do you mean by “stabilize” and “converge”? Please either provide definitions or plots to illustrate.
- Figure 7: Could be nice to include the dataset name in the caption.

---

> ### Author Response · Authors · 2024-11-20
> **Response to Reviewer Br4p**
>
> We thank the reviewer for the detailed comments. Below we note:
>
> **About missing details:**
> >1.  The paper misses dataset details (L195, L355, L427). Are NIAH samples created manually or by prompting large language models? What datasets are used to begin with in Sec 2-4? What additional evaluation tests did you create in Sec 4.1-4.3?
>
> > The paper misses experimental details, such as prompts used, links to existing assets used, etc.
>
> We appreciate the comment on missing details. Please see our **[Update on Experimental Details]**, where we have included comprehensive descriptions of the datasets, methods for creating NIAH samples, and evaluation tests. We would also be happy to provide further follow-up explanations. These updates will also be incorporated into the revised version of our paper. Our full reproducible code and data is open-sourced, and we will reveal the link after the anonymity period.
>
> **About $R$:**
>  > L156: By $R$, do you mean real numbers? If so, perhaps use $\mathbb{R}$ instead and clarify that  refers to unnormalized attention scores.
>
> Yes, $R$ refers to real numbers, and $\mathbb{R}$ is indeed more precise. We have updated the notation and clarified its meaning in the revised paper.
>
> **Question on Retrieval Head detection:**
> > L194: Does it happen that the model generates a word  that is not the next token in the needle  that should be generated? If this happens, do you skip the example? Or consider that as a case when all attention heads do not perform a copy-paste operation, even if an attention head actually pays the most attention to the token that should be generated next?
>
> - Our retrieval head detection is specifically targeting the copy-paste behavior. So yes, if the model generates a tokens that is not in the needle, we do not count it, no matter whether the attention head is attending to the target token or not.
> - That being said, since the needle sentence contain multiple tokens, to identify retrieval heads the model does not necessarily need to generate all of the tokens in the needle. Say if a needle "best place to visit in SF ..." contain 20 tokens, as long as the model copy-paste a fair portion of the tokens, say 14 out of 20, this level of copy-paste is enough for us the identify how strong/ frequently an attention head is doing retrieval.
>
>
> >L203: What do you mean by “stabilize” and “converge”? Please either provide definitions or plots to illustrate.
>
> - Since heads detected with strong retrieval scores in one context may not exhibit strong retrieval scores in another context, so we conduct retrieval head detection over multiple (question, answer pairs) and multiple long-contexts (haystacks).
> - As we increase the number of trials for detecting retrieval heads, “stabilize” and "converge” refers to the ranking of attention heads (based on retrieval scores) becoming consistent across repeated trials of retrieval head detection.
> - We have added definitions and included supporting plots in Section 7.2, Figure 13, in the updated Appendix.
>
> **Writings**
> > It’d facilitate reading to clarify that " is a sentence that is irrelevant to " in L146, instead of, for example, a short phrase or a single word. Can add a reference to Figure 2 so that readers see an example.
>
> >Figure 3: Seems that in fact less that 1% of attention heads are activated more than 50% times. The 5% in the caption could probably be changed to 1%.
>
> > Figure 7: Could be nice to include the dataset name in the caption.
>
> Thanks for the suggestions and we have modified the paper in in our new revision.

---

### Author Response · Authors · 2024-11-19
**[Update on Experiments] We also found Jamba has retrieval heads**

We are excited to observe that Jamba, a hybrid model combining **state-space layers (Mamba)**, mixture-of-experts (MoE), and a limited number of attention layers (4 layers × 32 heads = 128 total attention heads), also exhibits retrieval heads.
 Interestingly, these retrieval heads within the attention layers appear to play a key role in Jamba’s retrieval capabilities.
Below, we provide a comparison of **masking top-k retrieval heads**  versus **random heads** in  [Jamba-v0.1](https://huggingface.co/ai21labs/Jamba-v0.1)  (12B active parameters, with a total of 52B parameters across all experts) oon needle-in-a-haystack experiments using the same settings as Figure 7 in our paper:
| Masking Head number | 0   | 2    | 5    | 10   | 15   | 20   | 30   | 50   | 100  |
|-------------------|-----|------|------|------|------|------|------|------|------|
| Masking Random Head  | 100 | 99.1 | 98.0 | 94.7 | 90.3 | 85.1 | 70.3 | 44.1 | 9.0  |
| Masking Top Retrieval head| 100 | 98.1 | 38.5 | 61.9 | 33.0 | 15.3 | 10.3 | 12.9 | 3.5  |

These results provide strong evidence supporting our hypothesis that “full attention is crucial for effective long-context information retrieval” (L477) in the section “Relationship to Local and Linear Attention and State-Space Models”. Masking top retrieval heads significantly impacts retrieval performance for Jamba, confirming their essential role in maintaining model capabilities.
We will incorporate these updated results into Figure 7, along with comparisons to other models, to further strengthen our findings.

Please refer to Section 7.1 in the appendix to see the new figure with Jamba.

---

### Author Response · Authors · 2024-11-19
**Update on Experimental Details**

We thank all reviewers for their valuable feedback regarding the experimental details. In response to these comments, we have updated the paper to include the required details. Also we elaborate these information below:

### **1. NIAH experiments details:**
Before rebuttal:
- We manually construct / write four sets of (question, answer) pairs that are semantically irrelevant to the long document ("haystack"). The long documents for all four sets are randomly sampled from the publicly available [SlimPajama](https://huggingface.co/datasets/yaofu/slimpajama-per-source-length-upsample/viewer/default/train).
- We use three sets of (question, answer) pairs for retrieval head detection (i.e., calculating the retrieval scores discussed in Section 2), and reserve one set of (question, answer) pair for testing (generating Figure 1).
- A maximum sequence length of 50K tokens is used for retrieval head detection, while the full 128K token length is used during testing. This ensures that the retrieval head generalizes to longer lengths than detecting them.

The needles for retrieval head detection are:

1. Needle: A new report from the WMO shows that records were once again broken, and in some cases smashed, for greenhouse gas levels, surface temperatures, ocean heat and acidification. Question: What does a new report from WMO shows?
2. Needle: The best thing to do in Beijing is to take a walk in Chaoyang Park and have a cup of Espresso in the evening. Question: What is the best thing to do in Beijing?
3. Needle: Mr Green is disliked by everyone because he is a mean person and also he can't ride a horse or dive a car. Question: Why does everyone dislikes Mr Green?

During rebuttal:

Since reviewer ZPPp is concerned about the diversity of the needle, we use the above three cases as in-context examples to a language model, and ask the language model to generate 100 more examples. By doing this we get a synthetic data of 1.2K unique tokens spanning 10 topics (Technology, Transportation, Education, Festivals, Health .etc). Below are three examples of the synthetic data. The full needle set is uploaded as a supplementary material:

1. Needle: The newly constructed SkyBridge connects three cities—Everdale, Pinehurst, and Riverpoint—allowing citizens to travel with ease, admire scenic views, and save significant commute time. Question: What does the newly constructed SkyBridge connect and offer?
2. Needle: The upcoming GalaxyFest will feature over 100 sci-fi authors, 50 exclusive book signings, and a virtual reality experience of Mars colonization. Question: What features will the upcoming GalaxyFest have?
3. Needle: The Aurora Conservatory is renowned for its collection of rare Arctic flora, cutting-edge climate research, and eco-friendly glass dome architecture. Question: Why is the Aurora Conservatory renowned?

Repeating the retrieval head detection algorithm on Mistral 7B with our newly generated 100 needles, we get the same set of retrieval heads as before. This is to say, we confirm that out conclusion holds when scaling the number of needles from 3 to 100. We have added the details in appendix Figure 14.

We apologize for any confusion caused by the initial omission of these details and have updated the paper to include this information. Our full reproducible code and data is open-sourced, and we will reveal the link after the anonymity period.

### **2. Other Experiments and Prompts**
As our primary focus is the fundamental properties of retrieval heads, we do not emphasize prompt engineering, opting instead for the simplest possible prompts.
The datasets used, excluding **ExtractiveQA** (Details are mentioned in section 4.2 and we will open source it) and **NIAH**, are publicly available:
| **Dataset**       | **Prompt & Source**                                                                                                                  |
|--------------------|--------------------------------------------------------------------------------------------------------------------------------------|
| **Musique**        | Prompt and data from [LongBench](https://huggingface.co/datasets/yanbingzheng/LongBench). The CoT version adds: "please first think step by step." |
| **GSM8K and MMLU**    | Prompt and data from [Chain-of-thought-hub](https://github.com/FranxYao/chain-of-thought-hub).                                      |

We hope these clarifications address the reviewers' concerns

---

### Meta-Review · Area_Chair_Synt · 2024-12-16

**Metareview:**

The paper investigates how transformer-based models extract relevant information from long context. It identifies a specific type of attention head, named retrieval head, which plays a significant role in the information retrieval process. The authors demonstrate how they detect retrieval heads and describe their characteristics through various experimental settings. Additionally, they conducted experiments that involved pruning the retrieval heads to show that these heads are essential for recalling specific information amidst vast amounts of data. Their findings indicate that retrieval heads are crucial for extractive question-answering and chain-of-thought reasoning.

**Additional Comments On Reviewer Discussion:**

All reviewers agree on accepting this paper based on its soundness and novelty. Concerns have been generally addressed in the rebuttal.

---

### Decision · Program_Chairs · 2025-01-22

Accept (Oral)